# Dielectric Responses of Cytosolic Water Change with Aging of Circulating Red Blood Cells

**DOI:** 10.3390/cells14070486

**Published:** 2025-03-24

**Authors:** Larisa Latypova, Cindy Galindo, Leonid Livshits, Rodolfo Victor Teope, Dan Arbell, Gregory Barshtein, Anna Bogdanova, Yuri Feldman

**Affiliations:** 1School of Chemistry and Chemical Engineering, Harbin Institute of Technology, 92 West Da-Zhi Street, Harbin 150001, China; larisa.latypova@hit.edu.cn; 2Institute of Applied Physics, The Hebrew University of Jerusalem, Jerusalem 91904, Israel; cindy.galindo@mail.hujij.ac.il (C.G.); rodlfo.teope@mail.huji.ac.il (R.V.T.); yurif@mail.huji.ac.il (Y.F.); 3MIGAL Galilee Research Institute, Kiryat Shmona 11016, Israel; leonid.li@migal.org.il; 4Faculty of Sciences, Tel Hai Academic College, Upper Galilee, Kiryat Shmona 12110, Israel; 5Department of Pediatric Surgery, Hadassah University Hospital, Jerusalem 91120, Israel; arbell@hadassah.org.il; 6Department of Biochemistry, The Faculty of Medicine, The Hebrew University, Campus Ein Kerem, Jerusalem 91120, Israel; 7Red Blood Cell Research Group, Institute of Veterinary Physiology, University of Zurich, Winterthurerstrasse 260, CH-8057 Zurich, Switzerland; annab@access.uzh.ch

**Keywords:** microwave dielectric spectroscopy, red blood cells, Percoll gradient, hematological indices, RBC aging

## Abstract

Water molecules in the cytosol of red blood cells (RBCs) may exist in a free or bound state. The ratio between the free and bound water depends on the composition of the cytoplasm, particularly on the hemoglobin concentration. Microwave dielectric spectroscopy (MDS) provides information on the state of intracellular water in red blood cell suspension and the erythrocyte cytosol state. In the presented study, we used MDS to assess the differences in the free-to-bound water ratio in subpopulations of freshly donated human erythrocytes of different ages (young, mature, and senescent cells) obtained by fractionation in a Percoll density gradient. The obtained MDS parameters (dielectric strength ∆ε, the relaxation time τ, and the broadening parameter α) were compared with the red blood cell indices and single cell deformability measurements obtained for each subpopulation. We demonstrated that the unique hematological indices and deformability of red blood cells of different ages are well-correlated with the specific values of dielectric fitting parameters. The obtained results indicate that the dielectric properties of cytosolic water can serve as a sensitive marker of changes occurring in the cytosol of red blood cells during cell aging.

## 1. Introduction

While bound, water in cells forms an active matrix that affects the structure of the biomolecules, whereas free water allows diffusion and supports metabolic processes in living cells [1]. The interactions between the bound water and the host molecule (inorganic ions, lipids, sugars, and proteins) define the sub-molecular organization and dynamics of the water molecules within the hydration shell and the surface charges, structure, and function of the host molecule [2,3,4,5,6]. Our incomplete understanding of this interaction is caused mainly by the challenges in experimental characterization and quantification of the state of water and its properties in complex biological systems. Ruffle et al. [6] proposed that a large proportion of water in living cells should be in a state quite distinct from that of bulk extracellular water due to the high protein content of the cytosol. This difference is because protein concentration changes affect the water’s mobility and the hydration shell’s thickness [6]. Changes in the cytosolic protein concentration due to the movement of water and ions in or out of the cells are an additional factor that may alter the intracellular free-to-bound water ratio.

The hydration of highly abundant proteins, such as hemoglobin (Hb), has been shown to affect protein dynamics in a dose-dependent manner [7]. Changes in the mean cell hemoglobin concentration (MCHC) are associated with alterations in protein–protein and protein–water interactions, alterations in the hydration state of hemoglobin, and the dynamics of water within its hydration shell. The aging of RBCs was associated with changes in the MCHC, resulting from gradual dehydration and a decrease in the membrane surface-to-volume ratio [8].

The loss of cellular water and an increased MCHC make cells more rigid [9]. Thus, RBC dehydration leads to the deterioration of RBC deformability and, along with other factors, such as a reduction in membrane elasticity and stability, predisposing the cell to clearance in the spleen. Uncontrolled changes in the intracellular water content are signs of senescence. Most RBCs get dehydrated while aging, while a few of them undergo a terminal swelling phase [10,11]. The changes in the state of water in RBCs associated with their aging in circulation have never been addressed experimentally.

The present study focuses on assessing the free and bound water in the cytosol of RBCs forming low-, medium-, and high-density fractions that are enriched with “young”, “mature”, or senescent (“old”) cells, respectively [12]. MDS analyzed the state of the water in the cytosol of these RBC fractions. The cell age-dependent findings for the MDS spectral characteristics were then analyzed for their association with the RBC indices and cellular deformability.

The MDS technique allowed us to characterize cytosolic water’s structural and dynamic properties in intact RBCs. In addition, for each fraction, we provide common characteristics of the cell age, such as the RBC hematological indices and their deformability. The comparison between the dielectric data and RBC indices, together with deformability, confirms that the dielectric features of the erythrocyte intracellular water state are sensitive to in vivo RBC aging.

## 2. Materials and Methods

### 2.1. Experimental Design

To characterize the microwave dielectric parameters for cytosolic water in young, mature, and old fractions of RBCs, we fractionated freshly donated RBCs from six healthy subjects according to their density in a Percoll density gradient (see Appendix A). We determined the RBC indices, microwave dielectric spectrums, and deformability for the cells forming low-, medium-, and high-density fractions.

### 2.2. RBCs Preparation

Blood was obtained from four healthy donors at the Hadassah Hospital Blood Bank, following informed consent according to the Helsinki Committee Regulations Permit (0819-20-HMO, approval date 16 November 2020; Hadassah Hospital, Jerusalem, Israel). The blood was collected in standard test tubes containing citrate phosphate dextrose (CPD). A whole blood sample was fractionated in a Percoll-based gradient as previously described [13], with three fractions of the RBCs enriched with “young” (Y-fraction), “mature” (M-fraction), and “old” (O-fraction) cells in light, medium density, and dense subpopulations, respectively [14]. The cells were used for the DS measurements after three washes with 1x plasma-mimicking buffer supplemented with 0.1% BSA and 2 mM CaCl_2_. The control experiment was designed in the following way: after the Percoll separation, all the fractions were mixed, and the dielectric parameters were compared with the ones of the non-separated RBC sample. The control experiment was designed to determine whether cell properties change after the RBCs are separated into fractions. For this reason, we performed additional experiments where the characteristics of the native cells and those that underwent Percoll processing were compared. In Appendix A, we compared the hematological indices of the RBCs before and after fractionation in a Percoll gradient.

### 2.3. Dielectric Measurements

The dielectric measurements were carried out in a frequency range from 500 MHz to 40 GHz using the Microwave Network Analyzer (Keysight N5235B PNA-X), together with a Flexible Cable and a Performance Probe (Keysight N1501A Dielectric Probe Kit) [15,16]. The system was calibrated using three references: air, a Keysight standard short circuit, and pure water at 25 °C. The calibration was supported using the Ecal module. A special stand for the slim-form probe was designed and combined with a sample cell holder for liquids (∼7.8 mL). The holder was thermostabilized and attached to a Julabo CF 41 oil-based heat circulatory system. The temperature was maintained at 25 ± 0.1 °C. Each curve, corresponding to each fraction of the RBCs, was measured at least six times, and each measurement took 30 s. The real and imaginary parts, ε′(ω) and ε″(ω), were evaluated using the Keysight N1500A Materials Measurement Software 2018, with an accuracy of Δε′/ε′ = 0.05 and Δε″/ε″ = 0.05.

The dielectric relaxation peak of pure water at room temperature (25 °C) is localized in the microwave frequency band (0.5 to 50 GHz). The dielectric dispersion of water in biological systems is usually described using a combination of the phenomenological Cole–Cole (CC) equation and a conductivity term [17,18,19,20],(1)$$\varepsilon^{*\;}=\varepsilon_{\infty }+\frac{∆\varepsilon }{1+{\left(i\omega \tau \right)}^{\alpha }}-i\frac{\sigma }{\left(\;\omega \varepsilon_{0}\right)},$$

Here, $\varepsilon^{*\;}$ is the complex dielectric permittivity, ∆ε is the dielectric strength, $\varepsilon_{\infty }$ is the high-frequency limit of the dielectric permittivity, ω is the frequency, τ is the relaxation time, and α describes the dielectric loss peak broadening. In the case of pure water, α can be set to 1, resulting in the well-known Debye equation [21]. The parameter *σ* is the conductivity, and *ε*_0_ is the dielectric permittivity of the vacuum.

The microwave dielectric spectra of the RBC suspensions presented in Figure 1 reflect the relaxation process of water molecules in the cytoplasm and external medium [22]. Moreover, at the frequencies used in this research, the plasma membrane does not contribute to the dielectric response of the whole system; hence, the relative permittivity and conductivity of the cytoplasm can be calculated using a mixture equation.

We used a Kraszewski [23,24] mixture formula (Equation (2)) to evaluate the dielectric parameters of the cytoplasmic water for the RBC samples.(2)$$\varepsilon_{cyt\;}={\left(\frac{\varepsilon_{mix}^{\frac{1}{2}}-\varepsilon_{buff}^{\frac{1}{2}}\cdot \;\varphi_{buff}}{\varphi_{buff}}\right)}^{2},\;$$
where $\varepsilon_{cyt}$ is the dielectric permittivity of the cytoplasmic water, $\varepsilon_{mix\;}$—is the dielectric permittivity of the RBC suspension, $\varepsilon_{buff\;}$ is the dielectric permittivity of the buffer, and $\varphi_{buff}$ is the volume fraction of the RBCs.

### 2.4. RBC Deformability Measurements

The present research employed the computerized Cell Flow-Properties Analyzer (CFA), which we previously designed and constructed [25]. The CFA monitors the RBCs’ hemodynamic characteristics as a function of shear stress by directly visualizing their dynamic organization in a microfluidic (flow chamber) placed under a microscope. The deformability of the PRBCs (adhered to the glass slide) is determined by assessing their elongation under flow-induced shear stress [25].

In brief, 50 μL of the RBC suspension (1% hematocrit, in PBS, supplemented by 0.5% albumin) is inserted into the flow chamber (adjusted to 200 μm gap) containing an uncoated slide. The RBCs that adhere to the slide surface are then subjected to controllable flow-induced shear stress (3.0 Pa), and the change in cell shape is used to determine their deformability. This change is expressed by the elongation ratio, ER = *a*/*b*, where “*a*” is the major cellular axis and “*b*” is the minor cellular axis. ER = 1.0 reflects a round RBC, undeformed by the applied shear stress. The CFA contains an image analysis program capable of automatically measuring the ER for individual cells. The RBCs with an ER ≤ 1.1 are defined as “undeformable” (UDFCs), namely cells that do not deform under high shear stress. The image analysis produces an ER distribution (Figure 4) for a population of 10,000–15,000 cells, from which a series of deformability parameters can be derived, in addition to the median ER (MER) [25].

### 2.5. Characterization of Hematological Indices

XP-300 (Sysmex, Kobe, Japan) measured each sample’s standard hematological indices. We characterized the following parameters: the MCV (fL)—the mean corpuscular volume; the MCHC (g/dL)—the mean corpuscular hemoglobin concentration; and the MCH (pg)—the mean corpuscular hemoglobin.

## 3. Results

### 3.1. Spectral Characteristics of the Cells in the Fractions (Figure 1 and Figure 2 Modified with the Respective Controls and Parameters)

The typical dielectric spectrum of pure water is presented in Figure 1. This figure also shows the dielectric spectrum of phosphate buffer saline (PBS) and the young and old fractions of RBCs (15% vol. in PBS). Whenever water interacts with dipolar or ionic entities in aqueous solutions, the main dispersion peak (~20 GHz) broadens symmetrically and shifts to higher or lower frequencies. Other low-frequency dispersion processes may sometimes appear due to the solute contribution [26]. Accordingly, the direction of the shift is determined by the nature of the solute molecule: a shift to lower frequencies (red shift) corresponds to dipole–dipole interactions (non-polar solutes), and a shift to higher frequencies (blue shift) is typical of ion–dipole interactions [2,3]. These changes can be observed in the dielectric spectrum of the young and old RBC suspensions at 25 °C (Figure 1, orange and green lines). Figure 1Dielectric spectra of the young RBC suspension (orange line) and the old RBC suspension (green line) compared to phosphate buffered saline–PBS (red line) and distilled water (blue line). The real part of the static permittivity (ε′) is lower in the case of the RBC suspensions (area 1). Additionally, the imaginary part (ε″) of the RBC suspensions is characterized by the shift and broadening of the main relaxation peak (area 2) and the appearance of the conductivity tail (area 3). The hematocrit was 15% for both young and old RBC suspensions.
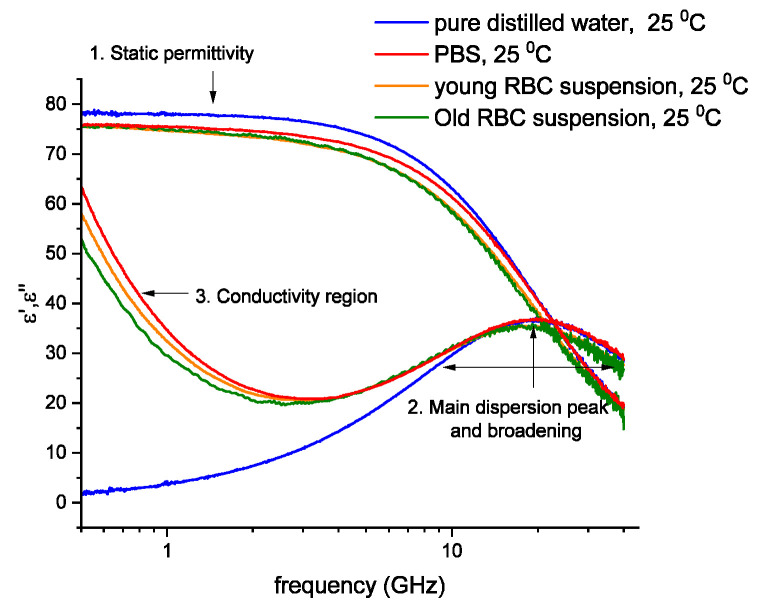


The interaction of the different components of the RBCs and molecules present in the medium with water results in changes in the dielectric spectrum of pure water. The main features include the reduction in the static dielectric permittivity (Figure 1, area 1), as well as the broadening and shift of the main dispersion peak (Figure 1, area 2), and the appearance of a conductivity tail (Figure 1, area 3).

Typical dielectric permittivity spectra for the cytoplasmic water of young and old RBCs, obtained using Equation (2), are presented in Figure 2. The state of bulk water in the cytosol can be revealed by assessing changes in the real part of the dielectric permittivity, ε′ (Figure 2A), as well as the broadening of the main dispersion peak (α-parameter) and the shift of its position (τ parameter) (Figure 2B).Figure 2(**A**) The real part of the dielectric permittivity (ε′) and (**B**) dielectric losses (ε″) of the cytoplasmic water of young RBCs (orange) and old RBCs (from the same donor). The cytosolic water was extracted using the Kraszewski mixture formula (see Equation (2)). The low-frequency regions in the dielectric loss spectra (**B**) are omitted, as they only represent ionic contributions. This study focuses on the state of bulk water in the system, which is characterized by the main dispersion peak at higher frequencies.
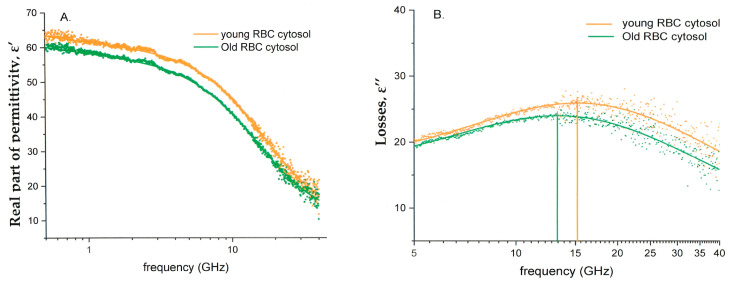


### 3.2. Does the Separation of RBCs in Percoll Cause Alteration in Cell Features?

As mentioned above, we have fractionated the RBC population into three subpopulations according to their density. The fractionation process in a Percoll density gradient involves exposing RBCs to both hydrostatic pressure (caused by centrifugation) and mechanical stress. At the same time, the cells pass through the Percoll-containing medium. Previously, we have shown that both shear and hydrostatic pressure, under certain conditions, may affect the deformability and aggregability of RBCs and the change in the structure of the membrane lipid bilayer [27,28]. In particular, we have demonstrated that the long-term exposure of cells to hydrostatic pressure created as a result of their centrifugation leads to a change in the viscosity of the lipid bilayer and an alteration in the RBC shape [27,28]. Therefore, in the first stage, a control experiment was designed to determine whether cell properties change after the fractionation of RBCs. In these experiments, the RBC hematological indices and dielectric fitting parameters (∆ε, τ, and α) of native cells were compared with those of the cells that underwent Percoll processing.

This comparison showed no influence of fractionation in a Percoll density gradient on either the RBC hematological indices (Appendix A) or the dielectric parameters reflecting the state of intracellular water (Appendix A). Based on these data, we can deduce that the exposure of the RBCs to Percoll and the fractionation process do not modify the properties of the RBCs intended for study in the fractionated form.

### 3.3. Hematological Indices of RBC in Fractions of Cells of Different Age

The selected hematological indices for RBC fractions are presented in Table 1. The fractionation of the RBCs allowed us to obtain RBC fractions with significantly different RBC indices (MCHC and MCV) but equal MCH values.

### 3.4. Dielectric Properties of Cytoplasmic Water

A comparison of the fitting parameters obtained after using a mixture formula Kraszewski (Equation (2)) for the cells forming the Y-, M-, and O-fractions obtained from the blood of four healthy donors is presented in Figure 3 and Appendix A.

The fitting parameters (the dielectric strength ∆ε, the relaxation time τ, and the broadening parameter α) differ between the fractions. Age dependence is most pronounced in the dielectric strength, ∆ε. The detailed analysis of correlations between the dielectric fitting parameters and RBC indices presented in Section 3.5 and Section 3.6 unravels some of the mechanisms behind the dielectric changes.

### 3.5. Deformability of Separated RBCs

For all the tested subpopulations of RBCs, we characterized their deformability, as demonstrated in Figure 4A. The elevation of RBC density is associated with decreasing RBC deformability, as expressed by the left shift of the elongation ratio (ER) distribution (Figure 4B).
Figure 4Deformability of RBCs from three density fractions. (**A**) Typical images of RBCs deformation and relevant distribution (**B**) of elongation ratio (ER) in RBCs population for cells from Y-, M-, and O-fractions. Shear stress is 3.0 Pa. Image size is 150 × 100 μm.
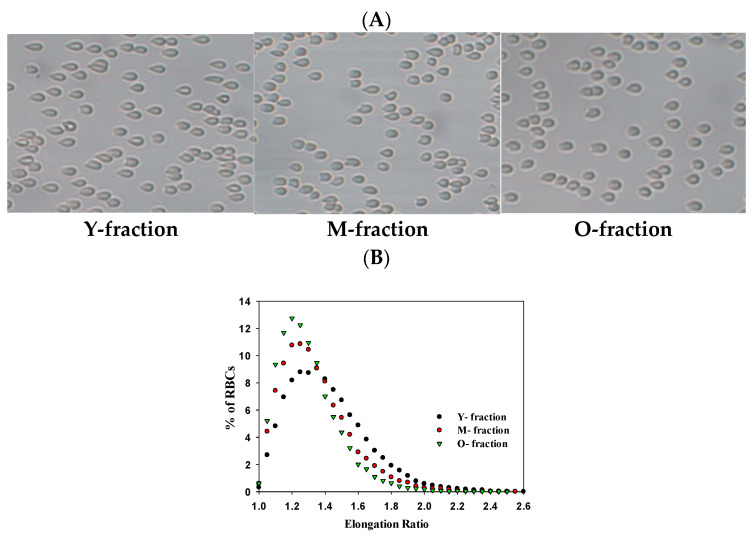



Previously we demonstrated [14] that the deformability of cells falls from young to mature to old RBCs. As shown in Figure 5, the value of the MER strongly decreased in this sequence. However, separating cells by density is not necessarily associated with their unambiguous partition by deformability. We have established [14] that the fraction of light (Y) cells includes a significant portion of undeformable cells with a low MCV, which fits the highest degree of heterogeneity in this fraction.

### 3.6. Dielectric Fitting Parameters of Cytosol Water vs. Hematological Indices

Figure 6 compares the RBC dielectric parameters with the MCHC for the RBC Y-, M-, and O-fractions. We also tested whether these dielectric fitting parameters were associated with the MCV (see Appendix A).

As follows from Figure 6, the dielectric strength (∆ε) shows the most robust age (density) dependence compared to the other parameters (τ and α), for which no correlation with cell density is observed. For this parameter (∆ε), a robust linear association with the MCHC and MCV could be detected (Figure 7).

### 3.7. Dielectric Fitting Parameters vs. RBC Deformability

The deformability of RBCs is expressed by two parameters: the median elongation ratio (MER) and the percent of undeformable cells (%UDFCs). It is essential to emphasize that in our previous publications [29], we demonstrated that %UDFCs is a feature that describes the percentage of aged cells in the RBC sample, while the MER assesses the averaged deformability of cells in the sample.

As shown in Figure 8 and Appendix A, we observed a correlation between the deformability markers and the dielectric characteristics (∆ε, α, and τ). The most significant correlation was observed for the dielectric strength (∆ε). A robust linear correlation (Figure 9) was detected between this parameter and both the average deformability (MER) and the concentration of undeformable RBCs (%UDFCs).

## 4. Discussion

The alteration in the microwave dielectric spectrum in the RBC suspension (see Figure 2) reflects the elevation of the MCHC value. This change, among others, may be associated with RBC aging processes, which leads to a decline in the RBC volume and membrane surface area [30], with a corresponding rise in the intracellular Hb concentration (Table 1) and a decrease in the intracellular electrolyte content and the osmotic water [31,32]. In our previous publication [15], we characterized the alteration in the dielectric fitting parameters Δε, α, and τ as a function of the methemoglobin (MetHb) concentration in the PBS buffer. We demonstrated that an increase in the MetHb concentration from 10 to 30 g/dL is associated with a reduction in ∆ε and α, and an increment in relaxation time (τ). The hemoglobin concentrations in the cytosol of the RBCs formed light, intermediate, and dense fraction ranges from 28 to 35 g/dL, increasing progressively with cell age (Table 1). Earlier on, we showed that the amount of free water in the MetHb solutions remained constant, while the concentration of metHb in the solution increased from 15 to 30 g/dL [33]. This phenomenon was explained by the cluster formation associated with releasing the water molecules from the hydration shells of Hb dissolved in the phosphate buffer. The self-organization of Hb molecules in solutions of Hb was also reported in other studies [34,35]. The present study shows this is not the case for the cytosolic Hb (Figure 6 and Figure 7).

An increase in the cellular hemoglobin concentration (MCHC) from 28 to 35 g/dL, due to dehydration and membrane loss (decrease in membrane surface area) with RBC aging, is associated with a progressive reduction in ∆ε. Thus, for the intact aging RBCs, an increase in cell density and increment in the MCHC is related to a decline in the amount of free water in the cells. It is also known that the solubility threshold for Hb in a buffered solution is 20 g/dL, while gel-like structures are formed at higher concentrations. In the cells, an MCHC of 34–36 g/dL is known to be a healthy reference range. Furthermore, heterogeneity in the hydration shells of hemoglobin molecules between the pre-membrane and cytosolic pools is eliminated in lysates, resulting in differences in spectral characteristics in the beta- and gamma-dispersion frequency ranges [36]. Furthermore, the difference in solubility may indicate that when in the buffer, MetHb molecules can form complexes [33], then in the cytosol, their formation by Hb is blocked due to the interaction of hemoglobin molecules with other cytosolic components, including proteins and other molecules and ions [37], thereby preventing the formation of complexes and ensuring a constant decrease in Δε with an increasing MCHC (Figure 7).

Our data show that, as it happens, part of the cytosolic water turns from a free to a bound state and becomes “invisible” to MDS. This phenomenon is best characterized by the shift in the dielectric strength value (Δε, see Figure 3 and Appendix A) [32,38]. The aging of RBCs is associated with a decrease in the Δε (Appendix A), suggesting an increase in the number of bound water molecules. The increase in the relaxation time (τ), along with the decrease in the dielectric strength Δε, and the broadening of parameter α with aging (Figure 2), indicates the more pronounced “red shift” (a shift in the water’s main relaxation peak to the lower frequencies) consistent with more dipole–dipole interactions of water dipoles with the cytoplasm components of the aging cells [32,38].

Previously, using Raman spectroscopy, several research groups studied the difference in the states of intracellular hemoglobin in young and old RBCs. The authors found that the intracellular hemoglobin in the old RBCs has some structural changes resulting in the dislocation of the tyrosine residues of the globins that may be detected as the alterations in Raman spectra at 830–850 cm^−1^ [39,40]. In contrast to the young RBCs, there may also be some denaturation, such as an increased abundance of unordered coil domains in the proteins [39,40]. The authors also note higher hemoglobin oxygenation levels in old than young cells [39,40]. Moreover, aging is associated with an increase in the pre-membrane hemoglobin pool [40], in which the hydration state of Hb molecules is most likely different from that in the cytosolic pool. The redistribution of the cytosolic water between the bulk and the bound pools is a cumulative reflection of all these processes.

Age-related changes in the cytosolic water state strongly correlate with RBC deformability features. The negative correlation between the relaxation time (τ) and deformability has been shown for cold-stored RBCs [16]. Deformability is a macroscopic physical property of the cell that depends on the properties of the cytosol (viscosity), cell surface/volume ratio, and membrane elasticity and stability [41]. The increase in the MCHC results from membrane loss during aging in vivo [41,42,43] and in vitro during storage [44,45]. The relaxation time (τ) is associated with the cytosolic viscosity, a function of the Hb concentration, the molecular conformation of Hb, and its ability to form sub-molecular aggregates. Looking at RBC aging through the changes in the state of cytosolic water allows us to associate it with the MCHC, MCV, and deformability, which serve as markers of RBC longevity.

In summary, we demonstrated that the aging of RBCs results in a shift in the balance between bound and bulk water, which may be detected using reflection in the microwave dielectric spectrum (see Figure 6). The elevation of RBC density, an increase in the MCHC, and a decrease in the MCV, which occurs as the cells age, is associated with the elevation of the ratio of bound to free water content (∆ε ↓), and the enhancement of the dipole–dipole type of interactions (τ ↑). Thus, the RBC age may be predicted for healthy human subjects based on the dielectric spectral characteristics.

## 5. Conclusions

The main conclusions are:Density separation using a standard set-up (based on the Percoll gradient) does not affect the dielectric parameters of cytosolic water.Microwave dielectric spectroscopy (MDS) can distinguish between subpopulations enriched with old, mature, and young RBCs (obtained by a Percoll gradient) due to the redistribution of bulk and bound water in the cytoplasm and the changes in its mobility.Dielectric parameters are associated with physiological properties of RBCs, such as deformability in age-separated fractions of cells.

## Figures and Tables

**Figure 3 cells-14-00486-f003:**
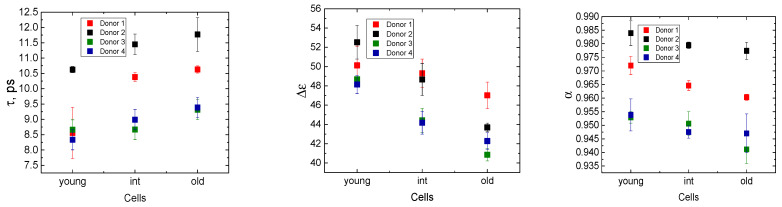
Fitting parameters for Y-, M-, and O-fractions of RBCs for four healthy donors. Each datum is Mean ± SE obtained from six measurements. Statistical analysis of data is presented in Appendix A).

**Figure 5 cells-14-00486-f005:**
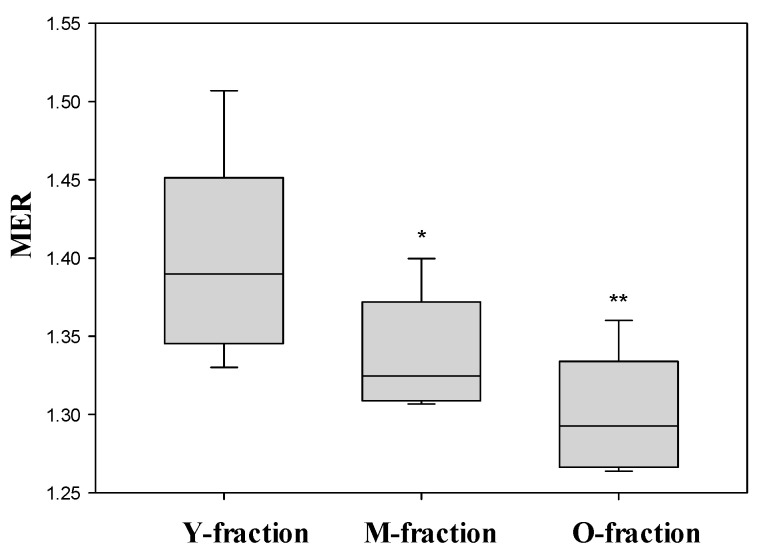
MER for the three density fractions (Y, M, and O) of erythrocytes. Each value represents the mean value ± standard error obtained from the measurements of six healthy donors. The significance of the difference in the MER value is shown for the M- and O-fractions compared to the young fraction. *p*—*, 0.01 and **, 0.0004.

**Figure 6 cells-14-00486-f006:**
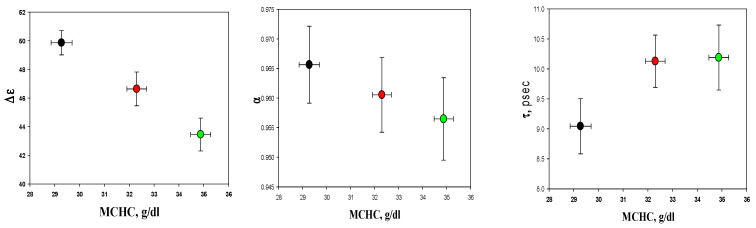
Correlation between dielectric fitting parameters and MCHC for three RBCs fractions: young (black), mature (red), and old (green). Each date was presented as mean ± SE from four samples.

**Figure 7 cells-14-00486-f007:**
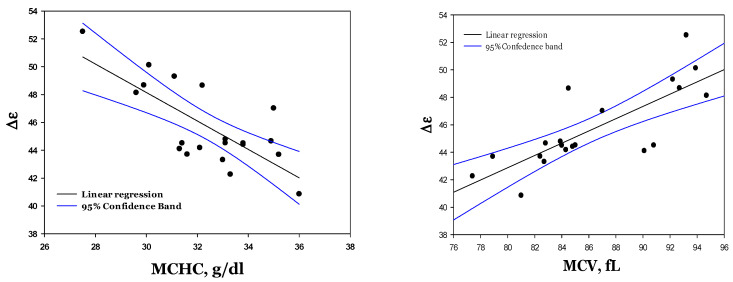
Correlation of the dielectric strength (∆ε) with the MCHC (left, R^2^ = 0.553; *p* = 0.00017) and the MCV (right, R^2^ = 0.613; *p* = 0.00005) for four healthy donors.

**Figure 8 cells-14-00486-f008:**
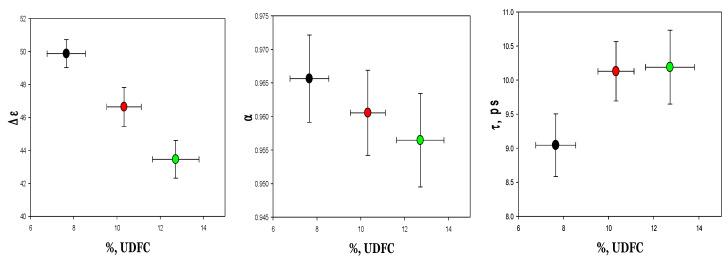
Correlation between dielectric fitting parameters and %UDFCs for three RBCs fractions: young (black), mature (red), and old fraction (green). Each date was presented as mean ± SE from four samples.

**Figure 9 cells-14-00486-f009:**
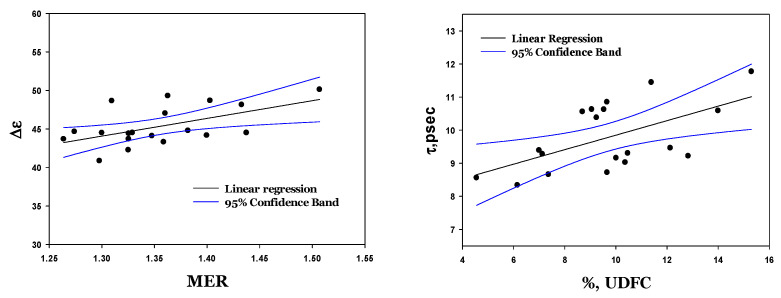
Correlation between dielectric strength (∆ε) with MER (left, R^2^ = 0.306; *p* = 0.014) and % UDFCs (right, R^2^ = 0.330; *p* = 0.01) for four healthy donors.

**Table 1 cells-14-00486-t001:** Hematological indices for three RBC subpopulations.

Parameters	Native		O-Fraction		M-Fraction		Y-Fraction
	Mean ± SE	Vs Y, *p* Value	Mean ± SE	Vs Y, *p* Value	Mean ± SE	Vs Y, *p*-Value	Mean ± SE
MCHC, g/dL	32.8 ± 0.4	0.002	35.0 ± 0.4	0.0001	32.5 ± 0.4	0.0012	28.9 ± 0.5
MCV, fL	86.3 ± 1.5	0.0025	81.4 ± 1.6	0.0002	86.6 ± 1.4	0.0015	94.7 ± 1.5
MCH, pg	28.0 ± 0.57	NS	28.5 ± 0.74	NS	28.1 ± 0.41	NS	27.4 ± 0.52

## Data Availability

Data are contained within the article. The data presented in this study are available upon request from the corresponding author.

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
