# Peer review of "Dielectric Responses of Cytosolic Water Change with Aging of Circulating Red Blood Cells"

_cells, 2025, doi:10.3390/cells14070486_

Round 1

Reviewer 1 Report

Comments and Suggestions for Authors

Using microwave dielectric spectroscopy (MDS) this interesting study detects differences in basic dielectric parameters between old and young red blood cells isolated from the high- and low-density region of a Percoll gradient, respectively. The differences correlate with changes in deformability and hematological indices between these cell fractions. These analyses and findings are novel and indicate that MDS is a valuable non-invasive method for investigations of the cytoplasm of cells.

However, prior to acceptance of this manuscript the following point (detailed below) must be carefully addressed/clarified by the authors:

Assuming that Figures 1 and 2 are derived from the same data set some inconsistencies between the graphs are conspicuous. The splitting between the curves of young and old RBCs seen in Figure 2 is not observed in Figure 1 apart from ε’’ in the conductivity region up to 4GHz.

Specifically,

·         within the static permittivity region in Figure 1 ε’ of old RBCs shows oscillations as compared to young erythrocytes and the peaks of these oscillations seem to have larger values than the respective ε’ values of young RBCs. Yet these features are not reflected in Figure 2A.

·         in the dispersion peak region there seems to be a nearly complete overlap between the curves of old and young RBCs in Figure 1, but there is a clear splitting in Figure 2B.

·         why is the 500MHz to 5Ghz region in Figure 2B skipped? This is the only region with a splitting in Figure 1. The respective effects of the transformation should be also shown for this region and discussed in the context of the authors’ hypothesis on cytoplasmic water.

Request for additional corrections/information 

·         Formula 2 is wrong: the denominator should contain the term (1-φbuff)

·         Table S5 rather than Table S3 should be referenced in lines 243, 408 and 409

·         Figure S1: Can the authors please provide information about the amounts of cells (as % fractions of total) isolated from the high and low density regions of the Percoll gradient that made up the O- and Y-fraction, respectively.

Author Response

Reviewer 1

Using microwave dielectric spectroscopy (MDS) this interesting study detects differences in basic dielectric parameters between old and young red blood cells isolated from the high- and low-density region of a Percoll gradient, respectively. The differences correlate with changes in deformability and hematological indices between these cell fractions. These analyses and findings are novel and indicate that MDS is a valuable non-invasive method for investigations of the cytoplasm of cells.

However, prior to acceptance of this manuscript the following point (detailed below) must be carefully addressed/clarified by the authors:

Assuming that Figures 1 and 2 are derived from the same data set some inconsistencies between the graphs are conspicuous. The splitting between the curves of young and old RBCs seen in Figure 2 is not observed in Figure 1 apart from ε’’ in the conductivity region up to 4GHz.

Specifically,

  1. within the static permittivity region in Figure 1 ε’ of old RBCs shows oscillations as compared to young erythrocytes and the peaks of these oscillations seem to have larger values than the respective ε’ values of young RBCs. Yet these features are not reflected in Figure 2A.

Within the dielectric permittivity region in Figure 1, ε’ of old RBCs exhibits oscillations compared to young erythrocytes, with the peaks of these oscillations appearing to have larger values than the corresponding ε’ values of young RBCs. However, these features are not reflected in Figure 2A.

It is important to note that 30 years ago, continuous spectrum measurements of this nature were not possible at all. Only with the advent of modern Vector Network Analyzers (VNAs) equipped with electrical calibration and open-ended coaxial probes has it become feasible to conduct such precise dielectric spectroscopy.

The high-frequency oscillations observed in the complex dielectric spectra arise from device noise rather than intrinsic properties of the samples. Consequently, an accurate spectra analysis is typically performed using a fitting function.

In our study, as we are primarily interested in the dielectric response of cytosolic water, we employed the mixture formula proposed by Kraszewski to eliminate the contribution of extracellular water pool into the dielectric permittivity and loss values.

Our analysis involves fitting the dielectric spectrum of cytosolic water (after applying the Kraszewski mixture equation) using two key components:

A conductivity term to account for ionic conductivity at low frequencies.

A Cole-Cole function is used to model the main dispersion peak, which appears around 20 GHz and corresponds to water's dielectric response.

The fitting equation used is:

This equation is included in the manuscript.

  1. in the dispersion peak region there seems to be a nearly complete overlap between the curves of old and young RBCs in Figure 1, but there is a clear splitting in Figure 2B.

The dielectric spectra in Figure 1 represent the dielectric response of water in the entire system, including both extracellular and intracellular compartments. In contrast, the spectra in Figures 2A and 2B depict the dielectric response of cytosolic water per averaged cell.

After applying the Kraszewski mixture formula, the contribution of extracellular water is eliminated, leaving only the signal corresponding to the intracellular water behavior. Despite noise, data analysis and fitting resolve the significant differences between the dielectric responses of old and young cells. We have included the explanations in the legend of Fig 2.

  1. why is the 500MHz to 5Ghz region in Figure 2B skipped? This is the only region with a splitting in Figure 1. The respective effects of the transformation should be also shown for this region and discussed in the context of the authors’ hypothesis on cytoplasmic water.

The conductivity region is irrelevant in this study, and we are focusing on the response of bulk water.

Request for additional corrections/information

  1. Formula 2 is wrong: the denominator should contain the term (1-φbuff)

We are grateful to the referee for this comment.  The Kraszewski formula used in our fittings is the one provided in the manuscript. Another variation, obtained through simple algebraic manipulation, is as follows:

To avoid confusion, we included the original form of the Kraszewski equation as it appears in his paper in the manuscript.

  1. Table S5 rather than Table S3 should be referenced in lines 243, 408 and 409

We appreciate the reviewer for highlighting this inaccuracy. Appropriate changes have been made to the text (see Lines 279, 303, 453, and 454).

  • Figure S1: Can the authors please provide information about the amounts of cells (as % fractions of total) isolated from the high- and low-density regions of the Percoll gradient that made up the O- and Y-fraction, respectively.

The relevant information has been added to the Fig. S1 legend.

Reviewer 2 Report

Comments and Suggestions for Authors

The manuscript investigates the dielectric properties of cytosolic water in human red blood cells (RBCs) and their changes with aging. Using Microwave Dielectric Spectroscopy (MDS), the authors assess the ratio of free-to-bound water in RBCs of different ages, fractionated by Percoll density gradients. The study establishes correlations between dielectric properties, hematological indices, and deformability of RBCs, suggesting that dielectric parameters can serve as markers for RBC aging The manuscript presents an innovative approach to studying RBC aging through dielectric spectroscopy. Despite its methodological strengths, the study would benefit from expanded controls, a larger sample size, clearer mechanistic insights, and a stronger emphasis on clinical applications. The supplementary data confirm the reliability of the experimental setup and fractionation method, reinforcing the findings. Addressing these points would enhance the manuscript's impact and applicability in research and clinical settings.

Comments: 

1. While a control experiment comparing fractionated and non-fractionated cells is included, the potential effects of fractionation stress on RBC properties require further clarification 

2. The study relies on RBCs from six healthy donors, which may not be sufficient to generalize the findings across a broader population.

3. The authors provide correlations but do not fully elucidate the mechanistic basis of changes in dielectric properties with RBC ageing.

4. Some figures and tables lack clear explanations of statistical analyses and error margins, making it difficult to assess data variability.

5. The study does not sufficiently discuss potential applications of dielectric measurements in clinical diagnostics or transfusion medicine.

Author Response

Reviewer 2

The manuscript investigates the dielectric properties of cytosolic water in human red blood cells (RBCs) and their changes with aging. Using Microwave Dielectric Spectroscopy (MDS), the authors assess the ratio of free-to-bound water in RBCs of different ages, fractionated by Percoll density gradients. The study establishes correlations between dielectric properties, hematological indices, and deformability of RBCs, suggesting that dielectric parameters can serve as markers for RBC aging The manuscript presents an innovative approach to studying RBC aging through dielectric spectroscopy. Despite its methodological strengths, the study would benefit from expanded controls, a larger sample size, clearer mechanistic insights, and a stronger emphasis on clinical applications. The supplementary data confirm the reliability of the experimental setup and fractionation method, reinforcing the findings. Addressing these points would enhance the manuscript's impact and applicability in research and clinical settings.

Comments: 

  1. While a control experiment comparing fractionated and non-fractionated cells is included, the potential effects of fractionation stress on RBC properties require further clarification.

 Thank you to the reviewer for the comment. We want to emphasize that we investigated the effect of the fractionation process on the hematological and dielectric indices of cells (section 3.2 and the relevant part in the Supplementary materials). We reached the same conclusion regarding cell deformability (Page 2; Supplementary materials). We have included an additional explanation in the revised manuscript (Lines 48 – 49 did not find the explanation in these lines, but it is probably somewhere there).

  1. The study relies on RBCs from six healthy donors, which may not be sufficient to generalize the findings across a broader population.

We appreciate the reviewer's question. As we show, the size of effect is sufficient to resolve the differences within a small cohort of 6 donors. This is a proof of principle study aiming to test the hypothesis of the possible association of RBC age, cell hydration, and the state of water in the cells. A more detailed study is required to address the possible differences in the state of cytosolic water associated with age and gender of healthy donors, as well as the possible changes associated with diseases. In line with the reviewer's suggestion, we have included a relevant remark in the "Discussion" section (see Lines 439-441).

  1. The authors provide correlations but do not fully elucidate the mechanistic basis of changes in dielectric properties with RBC ageing.

We appreciate the reviewer's question. The issue raised by the reviewer was critical to us during the preparation of the manuscript. We invested significant effort in establishing a potential cause-and-effect relationship between cell aging and variations in the dielectric parameters of the cytosol. In response to the reviewer's comment, we included the relevant text in the "Discussion" section (see Lines 447-451).

  1. Some figures and tables lack clear explanations of statistical analyses and error margins, making it difficult to assess data variability.

The revised manuscript (Lines 325 & 356) has been updated with the necessary corrections.

  1. The study does not sufficiently discuss potential applications of dielectric measurements in clinical diagnostics or transfusion medicine.

We appreciate the reviewer's comment. The revised manuscript (Lines 454 - 456) has been updated with the necessary corrections.

Reviewer 3 Report

Comments and Suggestions for Authors

This is a novel piece of work looking into the changes in the state of water in RBCs associated with their aging, with the dielectric and rigidity measurements. Since I am not familiar with the dielectric experiment, and I believe many clinical audients also not familiar with it, I hope the author putting more details. specifically:

1. please specify the equation that was used to fit the curves in figure 1. 

2. how the volume fractions of RBCs were determined?

3. in page 3, "Moreover, at the frequencies used in this research, the plasma membrane does not contribute to the dielectric response of the whole system". is there a reference or experimental evidence supporting this statement?

1. laxation time

Author Response

Reviewer 3

 This is a novel piece of work looking into the changes in the state of water in RBCs associated with their aging, with the dielectric and rigidity measurements. Since I am not familiar with the dielectric experiment, and I believe many clinical audients also not familiar with it, I hope the author putting more details. specifically:

  1. please specify the equation that was used to fit the curves in figure 1. 

Our analysis involves fitting the dielectric spectrum of cytosolic water (after applying the Kraszewski correction- Figure 2) using two key components:

  1. A conductivity term to account for ionic conductivity at low frequencies.
  2. A Cole-Cole function to model the main dispersion peak, which appears around 20 GHz and corresponds to the dielectric response of water.

The fitting equation used is:

Thanks for this handy comment. It is important to emphasize that in different complex systems, water can be considered the dipole subsystem, while other components may be defined as matrices. The morphology, dynamics, and dielectric properties of matrixes (porous materials, ionic and non-ionic aqueous solutions) are essentially different from those of water.

The broadening of dispersions linked to water in different complex systems can be described by the phenomenological Cole-Cole (CC) function:

 ,                                                                                 (1)

Where  is an empirical exponent describing the loss peak broadening. It was recently shown that whenever water interacts with another dipolar or charged entity, a symmetrical broadening of the dielectric loss peak of the main dispersion of the solvent occurs [1-5].

  1. how the volume fractions of RBCs were determined?

The XP-300 (Sysmex, Kobe, Japan) measured the hematocrit value. We added a relevant comment in the revised version of the manuscript (Line 197).

  1. in page 3, "Moreover, at the frequencies used in this research, the plasma membrane does not contribute to the dielectric response of the whole system". is there a reference or experimental evidence supporting this statement?

In reference 23 of the manuscript (Raicu et al. 2015, see (6) reference below), the lack of contribution of the RBC membrane to the dielectric response is shown for the frequency range we have used in our study.  At these frequencies [6],  the relative permittivity of the cytoplasm can be obtained from those of the cell suspension using an appropriate mixture equation. The optimal model in this case was first presented by Kraszewski [7], who considered microwave propagation in a heterogeneous medium. The underlying assumption is that a biphasic suspension can be viewed as a sum of an infinite number of thin water and substance layers, each of thickness δt << λ, where λ is the free-space wavelength. Consequently, the permittivity of the cellular interior can be represented as:

                                                  (2)

Here, e*mix, e*cell and e*buff are the dielectric permittivity of the mixture, the interior of the cell and the buffer, respectively. The volume fraction of the RBC (hematocrit) is represented by φ.

  1. Puzenko, A., P.B. Ishai, and Y. Feldman, Cole-Cole Broadening in Dielectric Relaxation and Strange Kinetics. Phys. Rev. Lett., 2010. 105: p. 037601-4.
  2. Levy, E., et al., Dielectric spectra broadening as the signature of dipole-matrix interaction. I. Water in nonionic solutions. J. Chem. Phys., 2012. 136: p. 114502-5.
  3. Levy, E., et al., Dielectric spectra broadening as the signature of dipole-matrix interaction. II. Water in ionic solutions. J. Chem. Phys., 2012. 136: p. 114503-6.
  4. Levy, E., et al., Dielectric spectra broadening as a signature for dipole-matrix interaction. IV. Water in amino acids solutions. Journal of Chemical Physics, 2014. 140(13).
  5. Latypova, L., et al., Dielectric spectra broadening as a signature for dipole-matrix interactions. V. Water in protein solutions. Journal of Chemical Physics, 2020.
  6. Raicu, V. and Y. Feldman, Dielectric Relaxation in Biological Systems: Physical Principles, Methods, and Applications. 2015, Oxford UK: Oxford University press. 432.
  7. Kraszewski, A., S. Kulinski, and M. Matuszewski, Dielectric properties and a model of biphase water suspension at 9.4 GHz. J.Appl.Phys., 1976. 47(4): p. 1275-1277.

Round 2

Reviewer 1 Report

Comments and Suggestions for Authors

The revised version of the manuscript includes now a more detailed description of the data analysis method which clearly improves the readability  of the manuscript.

The authors responded satisfyingly to all comments/suggestions raised by the reviewer except to the point that in Figure 2B there is only a reduced frequency range shown as compared to Figure 2A and Figure 1. 

This could make the reader a bit suspicious that there is maybe something to hide in the frequnecy range of 400MHz to 4GHz for the e'' as compared to the e' data. Possibly, the authors can explain their repective reasons shortly in the Figure legend.

Author Response

Thank you a lot.